# Fattening Iberian Pigs Indoors vs. Outdoors: Production Performance and Market Value

**DOI:** 10.3390/ani13030506

**Published:** 2023-01-31

**Authors:** Andrés Horrillo, Paula Gaspar, Ángel Muñoz, Miguel Escribano, Elena González

**Affiliations:** 1Department of Animal Production and Food Science, Faculty of Veterinary Medicine, University of Extremadura, 10003 Caceres, Spain; 2Department of Animal Production and Food Science, School of Agricultural Engineering, University of Extremadura, Avda. Adolfo Suarez, s/n, 06007 Badajoz, Spain

**Keywords:** Iberian pigs, technical-economic analysis, quality standard, farm profitability

## Abstract

**Simple Summary:**

This study attempts to analyse the technical and economic data of Iberian-breed pig farms that operate using two differentiated production models, i.e., the *cebo* or indoor fodder-fed pig production system where the animals are reared indoors, and the *cebo campo* or outdoor fodder-fed pig production system where the animals are reared outdoors with access to outdoor areas also during the fattening stage. The paper shows the study of three farms from each system with three batches per farm, where the data are obtained by way of a survey directly carried out with the farmers. Despite the conversion ratios being higher in the outdoor system, they do not translate into a significant increase in feed costs. However, labour costs are higher in the outdoor than in the indoor system. Gross margins are higher in the outdoor than in the indoor system, mainly due to the difference in incomes. Such incomes can vary temporarily subject to market prices. The outbreak of COVID-19 meant a sudden decrease in the gross profit, as the incomes went down and the costs remained at the same level. Crisis situations usually cause a reduction of the retail prices and a decrease in the profits.

**Abstract:**

The current Quality Standard regulating the Iberian pig provides for various differentiated farming systems subject to the type of management implemented and the breed of the pigs. This study attempts to analyse the differences between two of these production systems, i.e., the outdoor and the indoor rearing systems by comparing the main technical and economic factors of six farms, three operating under each system, in order to ascertain the most profitable production system. This analysis is based on the information provided by the farm owners. It also evaluates the impact that the COVID-19 pandemic outbreak had on profitability. The results show that both systems spend the same resources on animal feed, which represents nearly 60% of the expense, with the price of purchase of piglets representing 30–32% of the total; however, there are differences in the cost of labour, which is higher in the outdoor variant. In economic terms, outdoor farms obtained a higher gross margin than indoor farms did. Although their production costs are higher, these are offset with larger incomes due to the higher market price of the pigs at the time of slaughter. Lastly, all the farms under study reveal large financial losses on account of COVID-19, given that there was a general decrease in the revenues due to the decrease in the selling price of the pigs, which seems to be the most determinant factor for the economic profits made by these kinds of farms.

## 1. Introduction

Spain is the second producer of pork in the world, and this industry represents 39% of its Final Livestock Production [1]. Within the pork industry, the Iberian pig (autochthonous fatty pig breed that is traditionally produced in *dehesa* lands in the Southwest of the Iberian Peninsula) accounts for 6% of the total pork production [2]. Iberian pork products have improved their market position in recent years, increasing in importance and acceptance in national and international markets. In this sense, although pig production is very local, its export has significantly increased both in Europe [3] and in other countries such as Japan, Russia, and the USA, for example [4].

The higher demand of Iberian pork meat relies on its nutritional and sensorial qualities, which have been widely recognised for both fresh meat [5,6,7,8,9] and meat products, with the most popular product being the ham [10,11,12,13]. In addition to this fact, the current demand for meat products is not only due to nutritional and sensorial quality criteria but also other criteria associated with the various production systems and the feed type provided to the animals.

Consumers have increasingly become interested in the production methods employed to produce animal-derived products as they become more aware of the environment, the healthy attributes of food, and its production methods [14]. In this context, consumers associate certain products, and specially meat products, with an impact on the preservation of nature, and they tend to prefer the more sustainable systems [15] that produce ethically and can guarantee animal welfare [16].

It is common for meat consumers to associate animal welfare to outdoor animal production systems, that is, systems where the animals are reared outside the farming premises that invest energy resources on the maintenance of the animals and at times keep them in poor conditions. Ref. [17] highlighted that the majority of consumers believe that the Iberian pig production systems are less intensive and more environmentally friendly; however, reality tells us that the Iberian pig production system can be very varied.

The Quality Standard for Iberian meat, leg of ham, shoulder of ham, and loin [18] forms the current regulatory basis to produce Iberian pig. This standard sets out a labelling system based on the various production models. This allows for the rearing of Iberian pigs to offer a wider choice than extensive *dehesa* farms, where pigs are reared in free range conditions at their final fodder-feeing stage and their feed is based on acorn while grazing in *Montanera* [19], and also include intensive fodder-fed animal systems.

This fact becomes visible when we analyse the commercialisation of Iberian pigs, where out of the 3,754,704 Iberian-breed animals that were commercialised in Spain during the 2019–2020 campaign, 60% came from animals reared in intensive systems under the indoor *cebo* denomination, 20% were reared in extensive systems under the outdoor *cebo* de campo denomination, and the remaining 20% represented the acorn-fed bellota animals or animals that are finished in free-range and acorn-fed in the *dehesa* lands [20].

The various labels and Iberian pig production systems influence the pig selling price and the market price of its products [21], with the prices also having an impact on the profitability of the farms. In the current context, where the trend in meat demand is on the decrease, the value of the Iberian pig products must be maximized, especially when these products come from high-quality autochthonous animal breeds. Besides, these Iberian products have an added value provided by the positive perception consumers have of them as products derived from systems that look after animal welfare and sustainability, which makes consumers more willing to pay for their price [22].

In contrast, Iberian products, whether produced from acorn-fed, outdoor, or indoor animal systems are mainly commercialised through hotels and restaurants, which were badly impacted throughout 2020 on account of the restrictions imposed by the COVID-19 pandemic. Undoubtedly, situations like this are habitual within the cyclic economy that the Iberian pig market represents, but not all production models have been equally vulnerable to a pricing crisis as the one derived from COVID-19, and therefore, it is very important to be able to identify the systems that have proven more resilient. 

This study focuses on the evaluation of the production parameters and market value of the two majoritarian production systems, i.e., indoor and outdoor, within the Quality Standard for Iberian pigs. These two production systems have various characteristics that determine their production costs and varieties. Up to date, the studies focusing on the rearing and fattening of Iberian pigs in systems other than those finished in the *dehesa* traditional system are very scarce in terms of their technical and economic parameters. The currently available studies on Iberian pig intensive systems have focused either on a single system with the comparison of production and digestion parameters between Iberian and Landrace pigs [23] and/or the comparison between the effect of the sex of the animal, the castration of the females, and the weight at slaughter on the performance and the quality of the carcass in Iberian intensive management systems [24].

In this sense, given that the outdoor and indoor Iberian pig farms are two differentiated types of farms widely present throughout the Spanish territory, which produce the majority of the Iberian products consumed, it would seem necessary to have studies that further investigate their technical and economic characteristics as well as their potential association with the Iberian meat market and Iberian meat consumption. Based on this, the purpose of this study is to analyse Iberian pig farms through the main technical and economic performance pointers, and to evaluate the market differences according to the production system, i.e., outdoor or indoor. Lastly, the research finishes off with an analysis of the effects of the COVID-19 pandemic on these farms. Because of the appearance of the outbreak, there was a before and after in the market of Iberian products and another purpose has been to identify the effect of this outbreak on the farms, regardless of whether they were indoor or outdoor farms using the time of the appearance of the outbreak and the lockdown as a cut-off point.

## 2. Materials and Methods

### 2.1. Selection and Description of Farms and Systems

For the purposes of this study the data from six pig farms situated in the Southwest of the Iberian Peninsula were collected. The farms were selected through producer associations or personal contacts and were all certified under the Quality Standard for Iberian Pigs (*Real Decreto* 4/2014) with two types being selected: outdoor and indoor farms with animals of a 50-50 Iberian/Duroc cross breed. The farms specialised in the production of pigs only, so any farms that combined systems, included *Montanera*, or had reproduction pigs for self-supply of piglets were excluded from the selection. All the selected farms followed all in/all out production procedures. Additionally, for the ones under an outdoor system, the breeding and fattening of pigs had to comply with the requirements established in *Real Decreto* 1221/2009, which amends *Real Decreto* 1547/2004 and refers to semi-extensive indoor farms. Figure 1 makes a comparison between the requirements of the two production systems.

Information from three pig batches was collected from each farm (18 batches in total). The batches were produced sequentially and leaving a time for emptying and cleaning between batches of 15–25 days. Pigs are introduced to the farms at the approximate age of 3 months with a live weight of 26.42 ± 1.50 kg. The pigs were introduced to the various farms from the middle of 2017 to the middle of 2020. Pigs left the farms when they reached 165.90 ± 2.81 kg in live weight, when they were at least 10 to 12 months old, depending on the system, from the middle of 2018 to year 2021. Table 1 shows the main characteristics of the farms under study. 

Farms 1, 2, and 3 operate outdoor systems and have the largest areas; some of them include woodland. They have premises for the shelter of pigs, such as semi-open warehouses or barns. Two of them are provided with troughs and one of them spreads the food on the ground distributing it into purpose-made areas. All of them have water distribution systems with troughs. Those with electricity do not usually need it. Secondly, we have indoor pig farms, i.e., farms 4, 5, and 6, which are provided with smaller areas, and their premises are enclosed or semi-enclosed warehouses with concrete floors and outdoor patios for pig rearing and fattening. Generally, the warehouses are larger and more technological. The floors of the warehouses are made of prefabricated concrete grilled slabs for the collection of organic waste connected to septic tank. They are provided with electricity and the troughs operate automatically. The water supply is sourced from drilling wells within the farm, and water is distributed automatically.

### 2.2. Data Collection

Once the farms were selected, a farm visiting schedule was prepared for the collection of data by means of a survey. The survey collected information on the number of animals, weights and dates of entry and exit, weight in kilos, and type of fodder consumed as well as the number of deaths. Besides the study of the type of production, various economic pointers were obtained such as the price of the fodder, piglet price of purchase, price of sale of finished pigs, labour costs, electricity and water costs, veterinary service costs, medicines, and certifying agency costs. The prices of the animals and fodder were updated at the time of generating the expense or revenue. Based on all the variables collected through the survey, the various technical pointers were calculated such as the average daily feed intake, average daily weight gain and the conversion ratio, as well as financial data such as total income and cost and gross profit margin.

In addition to the type of production system (outdoor vs. indoor) with nine batches each, the 18 batches under study were divided into two significant time groups, i.e., batches produced before the start of COVID-19 lockdown (Pre-COVID batches) and batches produced after the start of the pandemic lockdown (COVID batches). A date for the animals to leave the farm to go to the slaughterhouse was selected, with 10 Pre-COVID batches and 8 COVID batches being analysed. For this reason, the number of pre and post COVID batches is not homogeneous, and there is no distinction between outdoor and indoor farms; therefore, analysis on the effect of COVID is based on timing, rather than making a comparison on farming systems.

### 2.3. Statistical Analysis

Statistical analyses were performed using an IBM SPSS Statistics software for Widows v 20.0 [25]. The results were provided using averages ± standard deviation (SD) for the two groups under study. The ANOVA test was performed for the identification of the differences between the indoor and the outdoor batches in terms of the technical and economic parameters calculated, as well as according to the time of sale being prior or after the COVID-19 pandemic outbreak. Technical parameters calculated from data collected were the initial body weight, final body weight, weight gain, stay days, total feed intake, average daily feed intake, feed intake, average daily gain, feed conversion ratio, mortality, adaptation feed, growth feed, and fattening feed. In the case of the economic parameters, these were feed cost, piglets cost, compensation of employees, medicines, certifying company, electricity, and fuel. The ANOVA test was chosen because the normality of the errors, as could be the case for some economic variables, has little influence on the F-contrast, since in the comparisons between means, these will always have a distribution close to normal according to the central limit theorem. Therefore, the results of these contrasts are substantially valid when the samples and observations are balanced, as is the case in this study.

### 2.4. Ethics Statement

This was an observational, prospective study of animals from commercial farms (Southwest of Spain), and no experimental interventions were performed. Data were recorded during the course of the habitual farming activities, without additional or invasive interventions required. Therefore, no ethical approval was required as stipulated in the Spanish Policy for Animal Protection, which complies with European Union Directive 2010/63/EU on the protection of animals used for research purposes.

## 3. Results

### 3.1. Outdoor vs. Indoor

Table 2 shows the production results from the comparison of the farms subject to their production systems.

The pigs from both systems enter the farm with a similar weight (26.42 ± 1.50 kg) and leave it when they reach 165.90 ± 2.81 kg in live weight, with no differences that are statistically significant in terms of weight or total weight gain (139.47 ± 3.65; *p* > 0.05). The difference between both systems lies in the age of slaughter, as age needs to be set at a minimum to comply with the standards. Outdoor pigs must be slaughtered when they are at least 12 months old and indoor pigs must at 10 months old. Both systems comply with the standards as pigs remain 310.11 and 237.33 days, respectively, for outdoor and indoor, in the finishing stage of fattening (*p* < 0.001), to which we must add 3 more months from the previous stage. Gaining the same weight during a different stretch of time is achieved through higher average daily gains in indoor animals (0.591 kg/d) than in outdoor animals (0.448 kg/d) (*p* < 0.001).

In order to achieve these daily weight gains and thus slaughter the animals at the adequate weight and age, each system must apply certain restrictions in the feed provided. Expressed as a percentage of live weight, the outdoor system provides 2.11% of live weight, whereas the indoor system is not so strict, providing 2.66% of live weight (*p* < 0.001). Although outdoor pigs daily ingest a smaller amount of fodder (2.02 kg) than indoor pigs do (2.57 kg) (*p* < 0.001), given that their feeding stage is longer, they end up consuming a higher amount of total fodder (627.73 vs. 609.62 kg) (*p* < 0.225). Consequently, the conversion ratios are worse for outdoor pigs (4.54) than for indoor pigs (4.37) (*p* < 0.041).

Mortality is a little higher in the outdoor system (2.78%) than in the indoor system (2.36%) (*p* < 0.244).

For the first week, pigs are fed on adapted feed (6–7% of the total) (*p* < 0.02), which is equivalent to the one they used to have in the previous stage and is progressively mixed with grower feed. Approximately, 50% to 54% of the fodder consumed is grower feed (*p* < 0.359). Final stage pigs in indoor systems feed on fattening feed, which represents 38% to 44% (*p* < 0.235) of the total feed ingested.

Table 3 shows the values of the economic variables obtained when comparing the production costs of the outdoor system against the indoor system. The results are expressed per pig produced.

In terms of production costs, the main difference is the labour costs of the production systems under analysis. The labour costs were 23.92 ± 0.38 €/pig for the outdoor system against 18.19 €/pig (*p* < 0.001) for the indoor system. Electricity and fuel costs are similar for both systems (0.79 €/pig and 0.82 €/pig *p* = 0.708) without significant differences between them (*p* > 0.708). In the same way, medicine costs, despite exceeding 0.10€ for outdoor pigs, do not reveal significant differences (*p* > 0.314). Both systems employ the same certifying company, and therefore, the cost for both is the same (*p* = 1.000).

The cost of fodder is 163.89 ± 5.78 €/pig in outdoor systems and 159.49 ± 9.18 €/pig in indoor systems, with a difference of 4.4 €/pig (*p* > 0.241). The cost of a piglet is similar in both farming systems, i.e., 84.96 €/pig for outdoor and €85.65/pig for indoor systems, with no significant differences (*p* > 0.760). These values will be subject to the time when the animals enter the farm and, consequently, to market prices, as expressed in Appendix A. In this sense, in this work, the data were analysed in terms of the real market price. These prices are not arbitrary and respond to the surcharge granted by consumer preferences due to the farming system, but what really fixes the prices are the production costs of each of the different farming systems.

Table 3 shows the distribution of the costs, with the feed costs being the highest expense for both farming systems, representing approximately 60%, followed by the cost of the piglets at 30.60% in the case of outdoor and 31.96% in the case of indoor systems. The remainder is distributed between the cost of labour employed and other operational costs. In terms of labour, the difference is statistically significant (*p* < 0.001), showing 8.60% in outdoor farms and 6.79% in indoor farms.

Table 4 shows the results from the economic balance of the two production systems. The results are shown as € per kg of produced pig, € per produced pig, and € per places produced pig.

In terms of the total cost, outdoor farming systems revealed 277.75 €/pig against indoor farming systems with 268.18 €/pig (*p* = 0.015). This meant a maximum difference of 9.57 €/pig between the two systems. However, the average gross margin in outdoor farms was 29.53 €/pig against indoor farms with 10.01 €/pig, with a maximum difference of 19.52 €/pig between systems. The profits per pig in the outdoor farming system exceeded those of the indoor system but were non-statistically significant (*p* = 0.304).

Therefore, the number of productive cycles possible in a year is 1.1 in outdoor systems and 1.43 in indoor systems. In this case, the time for emptying and cleaning between batches was 20.7 ± 7.85 days in outdoors systems and 18.2 ± 2.62 days in indoors systems. These balances were calculated per place that showed an average income of 397.30 €/place in indoors systems versus 338.19 €/place in outdoor systems (*p* = 0.012). This was a difference of 59.11 €/place. Similar results are shown for the average costs per place, with an average difference of 77.77 €/place between indoor vs. outdoor systems (*p* < 0.001). Therefore, the average gross margin results were 32.33 €/places in outdoors systems versus 13.67 €/place in indoor systems (*p* = 0.402), with the profit difference of 18.66 €/place not being significant between systems.

### 3.2. Effect of COVID-19 Lockdown on Iberian Pig Farms

Figure 2 shows the total revenues, costs, and margins for the farms before and during the COVID-19 pandemic. 

The average selling price before COVID-19 was set at 322.50 €/pig; however, during COVID-19, this went down to 255.54 €/pig with the difference being 67.0 €/pig (*p* < 0.001). However, the total costs of both production types are the same (*p* = 0.523). Sales prices affect the gross margin of both the Iberian outdoor and indoor pig farms. As Figure 2 shows, farms went from having high average profits of 48.30 €/pig to suffering considerable losses of −15.89 €/pig (*p* < 0.001).

## 4. Discussion

This study is an attempt to analyse the Iberian pig production systems employed by outdoor and indoor pig farms under the Quality Standard for Iberian Pigs requirements. The production systems are well known for the other categories of the Quality Standard (acorn-fed pig farms), their economic margins are stable, and consumers are perfectly familiarised with the product; however, the outdoor category does not follow the same patterns and is highly sensitive to economic and environmental factors that are extremely variable and can potentially become aggravated by a lack of definition of the Standard for these types of farms or products.

With this piece of research, we have tried to perform a case analysis to evaluate the two systems (outdoor and indoor) from a technical and economic viewpoint and how these factors influence their prospects and on the industry’s ability to pass these parameters on to the consumer. 

From a technical viewpoint, the low growth capacity of the Iberian breed [26] is improved when it is crossed with the Duroc breed, obtaining a higher growth potential [27]. Ad libitum feeding allows for this growth potential to materialise through an adequate weight being reached at an earlier age of slaughter (8 months) [27,28]. When we extend the age of slaughter to 10 and 12 months old, reducing the Average daily feed intake (ADFI) and therefore reducing the Average daily gain (ADG) [28], total consumption is increased for the entire period with the resulting decrease in the Feed conversion ratio (FCR). Similar data are obtained in our study, with the outdoor animals being slaughtered at 12 months old with higher FCR than indoor animals at 10 months of age. Such differences can be mainly due to the use of the energy they ingest other than for growing purposes as the feed provided for growing the animals is the same in both cases.

In the case of the outdoor system, which has a growing period of two more months than the indoor system, there is an increase in the global energy costs, mainly due to animal maintenance requirements that can translate, subject to the system management, into an increase in costs. In addition to this, in this type of production, there is an increase in the physical activity of the outdoor reared pigs as they are provided with larger areas to graze during their final fattening stage (100 m^2^ vs. 2 m^2^), and this increases the need for supplementary energy [29]. Pigs are also be exposed to weather conditions that may increase their metabolic energy requirements, either on account of high or low temperatures [30]. The weather conditions of the pig’s breeding areas (Southwest of Spain) with Mediterranean climate and Atlantic influence means that the animals are subjected to low temperatures in the winter and high temperatures in the summer.

However, when the results of the economic balance are shown by productive cycles on the farm and by number of places on the farm, the longer time of the outdoor pigs is not a decisive factor. In indoor systems, the total income is higher than that obtained in outdoor systems in a year.

In contrast, the higher degree of mortality of the outdoor animals also implies a worse conversion ratio. The older the animal at the time of death, the more prominent that increase will be, as they will have production diseases, such as locomotion issues: indoor reared vs. outdoor reared [31]. The pigs have a quite satisfactory health status as they adhere to the sanitary procedures of vaccinations and preventive parasite control. This is reflected in the low mortality rate compared to the average observed in Spanish farms [32].

Given the above notions, in the case of outdoor reared animals, the requirements would technically be more demanding than those of the indoor system and that these characteristics could have an impact on the final production costs of these types of farms. As described, the technical parameters have an impact, but also the fact that the outdoor pig category includes a very wide range of farms and management systems, with animals that have been bred to be Iberian acorn-fed but could not be certified in the end, to animals that are bred and reproduced under a system that is similar to the indoor system.

Several authors highlight that, currently, the production systems available for the outdoor category and the acorn-fed category of pigs are concurrent, necessary, and complementary [17,21]. Although the acorn-fed animal feeds on the natural resources available in the *dehesa* land during the *Montanera* period, which makes this livestock production system more environmentally friendly, the outdoor fodder-fed animal system allows for the fattening stage to take place when there is no acorn available due to seasonal restrictions. The combined use of the *Montanera* system for the fattening of pigs and outdoor fodder feeding is the optimum fattening strategy to improve the sustainability of the traditional production of Iberian pigs [33].

The analysis of the farm economic management has revealed that the main difference found in terms of costs between both types of farms was 9.57 €/pig, and this difference was mainly due to labour costs. An assessment of this factor highlights a fact that was mentioned earlier on, i.e., the standard regulating the Iberian production industry establishes the minimum age for slaughter of outdoor pigs at 12 months, which has a linear effect on the increase of the labour costs in those farms. Similar results are obtained by the research work carried out by [21], which compares the costs of labour between Iberian pig intensive (30 €/pig) and extensive (41 €/pig) systems, identifying a more prominent difference between indoor animal farms (intensive) and acorn-fed animal farms (extensive), since the acorn-fed pigs, named de bellota according to the currently-effective standards, cannot be slaughtered before they are 14 months old.

The profit margins obtained in the outdoor pig system have been noticed to be higher than those for indoor pig systems, but careful attention must be paid when taking them into account as the results have historically been cyclical and can influenced by numerous factors [34,35]. In this sense, it is common to find a seasonal pattern in the prices of piglets, with the lowest prices being found in the summer and the highest prices in the winter, whereas the fattening animals have their lowest prices in the autumn and the highest in the winter [36]. (Appendix A.)

A recent example of the market price fluctuation (Appendix A) and the changes in the consumption patterns was the COVID-19 pandemic, which brought about a decrease in the market consumption of higher-value products due to its economic consequences on consumer expense. In the case of meat companies, and particularly, the Iberian pig subsector, this situation had a special impact on the most exclusive and pricy products in the market. One of the reasons was largely caused by the closure of hotels and restaurants. Studies such as that of [37] revealed the negative effects endured by companies within the entire value chain of the Iberian pork products with a decrease in their sales.

Currently, we are still waiting to see the impact that the general increase in the cost of raw materials and fuel will have on the margins and retail price of these products.

Outdoor system obtains a higher gross margin than indoor farms, as shown in Appendix A. Although their production costs are higher, these are offset with larger incomes due to the higher market price of the pigs at the time of slaughter (Appendix A).

One thing remains clear, and this is that meat products with differentiated characteristics such as the outdoor pork can have an influence on the consumer and on retail price [38]. However, a large proportion of the consumer population is still unfamiliar with certain attributes [39] of the outdoor system, such as the difference in the production model to the indoor system. These aspects that make the outdoor pig system different from others must be promoted if their producers are to benefit, since the impact of food production on the natural resources, climate change, and animal welfare is causing a decrease in the consumption of meat and meat products [40,41]. In contrast, the search for healthy products [42,43] can also play a beneficial part in these types of production farms.

## 5. Conclusions

As we have attempted to demonstrate, not all the Iberian pig production systems are the same nor are all their differentiating aspects as well known, despite them being regulated by the Quality Standard for Iberian Pigs. There is a general lack of awareness about the operation of outdoor and indoor farming systems, which is partially the result of the outdoor system always being surrounded by a certain level of instability deriving from its lack of positioning within the standard. There are also scarce bibliographical references available both on the technical and economic aspects of these production systems.

The higher production costs involved in the outdoor system have a long-term influence on its profit margins, specially within the context of the dramatic increase of the price of the raw materials. These types of outdoor production systems are appreciated by consumers, but it is necessary to promote their positive qualities, and perhaps, there is a need to have a more differentiated regulation against the indoor systems, if the environment and animal welfare are to be improved as well as if a fair price is to be paid to the producers.

In this sense, the market price of Iberian pig production is always marked by the price at which the animals enter the market and the final demand for the product. These prices are not arbitrary and respond to the surcharge granted by consumer preferences due to the farming system and the production costs of each of the different farming systems.

## Figures and Tables

**Figure 1 animals-13-00506-f001:**
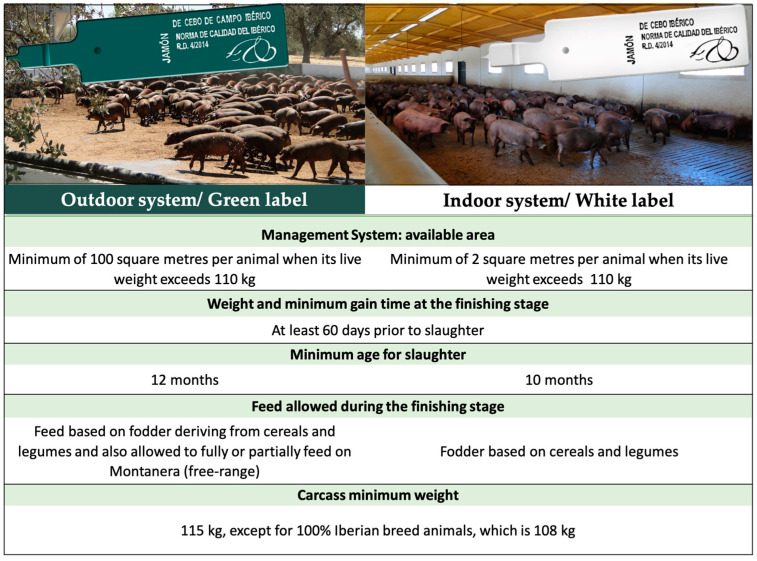
Requirements of the production characteristics of outdoor and indoor Iberian category products according to the currently valid Quality Standard for Iberian Pigs.

**Figure 2 animals-13-00506-f002:**
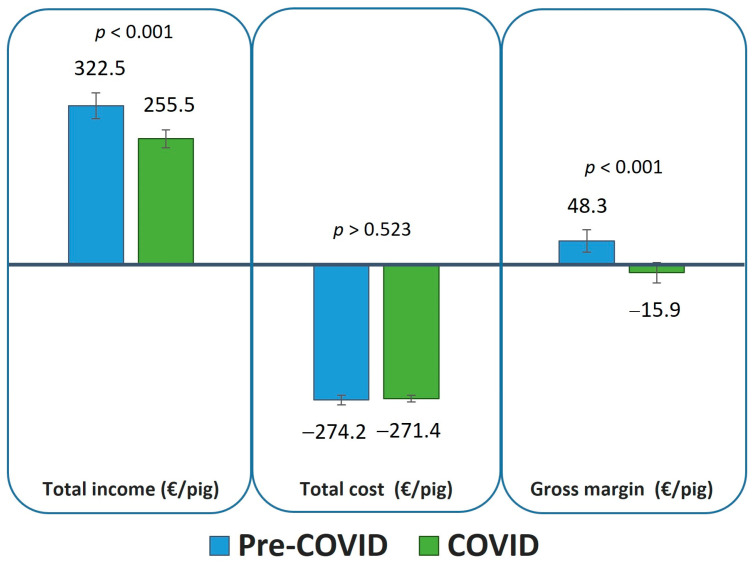
Comparison between the revenues, costs, and gross margin before (Pre-COVID) and after (COVID) the start of lockdown (March 2020) on account of the COVID-19 pandemic restrictions.

**Table 1 animals-13-00506-t001:** Farm characteristics.

	Outdoor			Indoor		
Item	Farm 1	Farm 2	Farm 3	Farm 4	Farm 5	Farm 6
Capacity/No. of places (pig)	400	350	500	800	520	700
Total area (ha)	20	11	90	4	6	3
Wooded area (ha)	Yes	No	Yes	No	No	No
Livestock infrastructures (m^2^)	375	216	400	1680	350	600
Feed storage structures “silos” (kg)	12,000	No	15,000	48,000	8000	30,000
Feed troughs	Yes	No	Yes	Yes	Yes	Yes
Water system	Yes	Yes	Yes	Yes	Yes	Yes
Electricity	Yes	No	Yes	Yes	Yes	Yes

**Table 2 animals-13-00506-t002:** Growth performance results: outdoor vs. indoor.

	Outdoor		Indoor		
Item	Average	SD	Average	SD	*p*-Value
Initial Body Weight, (kg)	26.25	1.02	26.59	1.92	0.591
Final Body Weight, (kg)	165.27	1.02	166.53	3.86	0.357
Weight Gain, (kg)	139.01	1.54	139.93	5.04	0.620
Stay Days, (d)	310.11	5.25	237.33	9.49	0.000
Total Feed Intake, (kg)	627.73	20.98	609.62	37.59	0.225
Average Daily Feed Intake, (kg/d)	2.02	0.08	2.57	0.21	0.000
Feed Intake, (% BW)	2.11	0.09	2.66	0.19	0.000
Average Daily Gain, (kg/d)	0.448	0.009	0.591	0.036	0.000
Feed Conversion Ratio, (kg/kg)	4.54	0.17	4.37	0.15	0.041
Mortality, (%)	2.78	0.56	2.36	0.89	0.244
Distribution of Feed Types %					
Adaptation Feed, (%)	7.40	0.86	5.80	1.63	0.019
Growth Feed, (%)	54.03	7.57	50.44	8.49	0.359
Fattening Feed, (%)	38.58	7.54	43.75	10.07	0.235

SD: Standard deviation; BW: Body weight.

**Table 3 animals-13-00506-t003:** Comparison of cost pointers: outdoor vs. indoor.

	Outdoor		Indoor		
Item	Average	SD	Average	SD	*p*-Value
Feed Cost (€/pig)	163.89	5.78	159.49	9.18	0.241
Piglets Cost (€/pig)	84.96	3.49	85.65	5.62	0.760
Compensation of Employees (€/pig)	23.92	0.38	18.19	0.74	0.000
Medicines (€/pig)	3.27	0.41	3.12	0.18	0.314
Certifying Company (€/pig)	0.90	0.00	0.90	0.00	1.000
Electricity and Fuel (€/pig)	0.79	0.15	0.82	0.22	0.708
Cost distribution					
Feed Cost (%)	59.00	1.05	59.45	2.36	0.603
Piglets Cost (%)	30.60	1.04	31.96	2.18	0.112
Compensation of Employees (%)	8.60	0.26	6.79	0.36	0.000
Other Expenditure (%)	1.81	0.14	1.82	0.15	0.871

**Table 4 animals-13-00506-t004:** Comparison between selling prices and economic balance: outdoor vs. indoor system.

		Total Income			Total Cost			Gross Profit		
		Average	SD	*p*-Value	Average	SD	*p*-Value	Average	SD	*p*-Value
€/kg	Outdoor	1.86	0.31	0.115	1.68	0.05	0.001	0.18	0.29	0.304
	Indoor	1.67	0.15		1.61	0.05		0.06	0.15	
€/pig	Outdoor	307.28	50.31	0.14	277.75	7.26	0.015	29.53	49.53	0.308
	Indoor	278.19	25.01		268.18	7.65		10.01	25.31	
€/place	Outdoor	338.19	54.45	0.012	305.86	11.36	0.000	32.33	53.88	0.402
	Indoor	397.30	30.53		383.63	17.31		13.67	36.50	

SD: Standard deviation.

## Data Availability

This is not applicable, as the data are not in any data repository with public access. However, if an editorial committee needs access, we will happily provide them with it.

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
