# Peer review of "Fattening Iberian Pigs Indoors vs. Outdoors: Production Performance and Market Value"

_animals, 2023, doi:10.3390/ani13030506_

Round 1

Reviewer 1 Report

Author investigates the indoor and outdoor system for Iberian pigs before and after pandemic. 

The statistics analysis should be revisited. The factor such as the days at farm should be considered including in equation. Moreover the results and discuss post pandemic between two production system is missing.  

Line 131 With all in all out system, how are those slow growing pigs are handled when carcass minimum weight is 115 kg?

 Line 166 As indicated above there were three batches from each production site, why there is 10 pre Covid group and 8 Covid group? For 8 Covid group, how many of groups with piglets purchased after Covid lockdown?

Line 173: Student t test isn’t the right tool for economic data analysis.

Table 2 The decimal place of P value should be adjusted to where number can be read

How is the stay days determined? Was it based on the very last pig reached 115 kg?

Line 209 I believe the P value is incorrect? Please also leave the space when come to report the P – Value (P < 0.244 instead of P<0.244).

Please also provide the health status of farms that selected for the study.

 Line 226 It should be P = 0.708?

Line 231 missing “P” for = 0.241.

Table 4. There is one more thing that need be considered in this equation is number of pigs can be produced per year in each type of farming system.  

 Given the fact that market price is fluctuate year long. With two months different the profit can be quite different. How are market price treated in the equation?

Line 250 Perhaps this is not significantly different 

Figure 2. Are there any different between outdoor and indoor production system during pandemic?

That is one of objective for this article?

Line 280 Pigs should eat more feed when they grow older. Why ADFI can reduce when extend age f slaughter from 8 months to 10/12 months?

Line 287 Therefore Days of occupation at farm should be include in equation when discuss the profit of production.

Author Response

Response to Reviewer 1 Comments

Author investigates the indoor and outdoor system for Iberian pigs before and after pandemic.

The statistics analysis should be revisited. The factor such as the days at farm should be considered including in equation. Moreover the results and discuss post pandemic between two production system is missing. 

Following the reviewer's comments, the statistical analysis has been revised and the number of days has been added to the equation for the analysis of pig systems. These results are shown in the text both in the methodology part and in the results, Table 4.

Line 131 With all in all out system, how are those slow growing pigs are handled when carcass minimum weight is 115 kg?

In order not to be rejected at the slaughterhouse for having carcasses of less than 115 kg, and thus falling out of the Quality Standard, the pigs do not leave the farms with less than 145 kg live weight, since the carcass yield of this type of production is usually 80%. The pigs that make up the batches that are sent to slaughterhouse have a lower limit of this weight, the average being 165 kg (table 2). Generally, farms make prospective weights before sending a batch to slaughter.

 Line 166 As indicated above there were three batches from each production site, why there is 10 pre Covid group and 8 Covid group? For 8 Covid group, how many of groups with piglets purchased after Covid lockdown?

The primary objective of the work was a comparative analysis of the two systems (outdoor-indoor) but in the period of data collection, the COVID-19 pandemic appeared. The analysis of the effect of COVID on the study farms took place a posteriori and is not part of the experimental design. 

As a consequence of the appearance of the outbreak there was a before and after in the market of Iberian products and, in this study, we have studied the effect of this outbreak on the farms, regardless of whether they were indoor or outdoor farms. Therefore, the batches that have been compared 10 and 8 have been established using the time of the appearance of the outbreak and the lockdown as a cut-off point.

For this reason, the number of pre and post COVID batches is not homogeneous and there is no distinction between outdoor and indoor farms and therefore the conclusions of the effect of COVID cannot be given in relation to the production system. 

A brief clarification has been introduced in the text in order to allow the reader to identify that the objective of the analysis of the COVID effect is not linked to the production system.

Line 173: Student t test isn’t the right tool for economic data analysis.

Some preliminary analysis was cited in error in the text. The data in the paper were analyzed by analysis of variance using the ANOVA model, since this is a robust technique. This test was chosen because the normality of the errors, as could be the case for some economic variables, has little influence on the F-contrast, since in the comparisons between means these will always have a distribution close to normal according to the central limit theorem. Therefore, the results of these contrasts are substantially valid when the samples and observations are balanced, as is the case in this study.

At the same time, the text of the article has been corrected.

Table 2 The decimal place of P value should be adjusted to where number can be read

Corrected.

How is the stay days determined? Was it based on the very last pig reached 115 kg?

It is calculated as the number of days elapsed between the entrance of the first group of pigs in the farm and the exit of the last group of pigs sent to the slaughter- house.

Line 209 I believe the P value is incorrect? Please also leave the space when come to report the P – Value (P < 0.244 instead of P<0.244).

Corrected.

Please also provide the health status of farms that selected for the study.

The pigs have a quite satisfactory health status. The sanitary procedures of vaccinations and preventive parasite control for these productions are applied to them.

This is reflected in the low mortality rate (table 2). The health status of the pigs is included in the discussion.

 Line 226 It should be P = 0.708?

Corrected

Line 231 missing “P” for = 0.241.

Corrected

Table 4. There is one more thing that need be considered in this equation is number of pigs can be produced per year in each type of farming system.  Given the fact that market price is fluctuate year long. With two months different the profit can be quite different. How are market price treated in the equation?

Following the reviewer's indications, a new section has been added to Table 4, showing the average results per production cycle for each system. In this calculation, both time and number of pigs per cycle were taken into account.

Line 250 Perhaps this is not significantly different

Following the reviewer's comments, the results have been revised and a sentence has been added to the text

Figure 2. Are there any different between outdoor and indoor production system during pandemic? That is one of objective for this article?

As I have already mentioned regarding one of your previous comments, the primary objective of the work was a comparative analysis of the two systems (outdoor-indoor) but in the period of data collection, the COVID-19 pandemic appeared. The analysis of the effect of COVID on the study farms took place a posteriori and is not part of the experimental design. Therefore, it was not one of the objectives of the study.

Line 280 Pigs should eat more feed when they grow older. Why ADFI can reduce when extend age f slaughter from 8 months to 10/12 months?

It is true that animals feed consumption increases as live weight increases and, in our study, the same happens, since a progressive feeding programme is applied.

However, to increase the fattening days and to reach equal live weight in 10 and 12 months of age, for the indoor and outdoor groups respectively, a feed restriction process is required, in which a smaller amount of feed per day is provided in outdoor group.

The two months longer fattening period in the outdoor group increases the feed expenditure for maintenance and therefore the total feed consumption over the whole period, which results in poorer feed conversion rates.

This has already been explained in the discussion.

Line 287 Therefore Days of occupation at farm should be include in equation when discuss the profit of production.

Following the reviewer's comments, days of occupation at farm has been included in the equation for the systems analysis. A paragraph has also been included in the text.

Reviewer 2 Report

Animals manuscript: Fattening Iberian pigs indoors vs outdoors: production performance and market value by Andrés Horrillo, Paula Gaspar, Ángel Muñoz, Miguel Escribano and Elena González is well written and I have very little issue with the paper. However, one main point I am unable to discern is the difference in gross revenue between indoor and outdoor production. Are the economics of the two systems being compared with an identical market value, or are the differences in value simply differences in market price fluctuation? I do not know much of the Iberian marketing system, but are indoor and outdoor pigs paid the same, or is there a difference with outdoor pigs given a premium because they are outdoor raised?

It would seem to me that you would want to compare the two systems on an equal market price per kg of pork. If this is the case, then I think the data would suggest an economic advantage to indoor-raised pigs.  If there are premiums to outdoor raised pigs, then this should be added on top of an identical market price. From reading the paper, I can not determine if the systems are being compared on an equal market value basis, or just the arbitrary market prices received at the time pigs happened to be slaughtered.

I think the economic aspects of the paper need to be explained and prices equalized before consideration for publication.

Author Response

Response to Reviewer 2 Comments

Animals manuscript: Fattening Iberian pigs indoors vs outdoors: production performance and market value by Andrés Horrillo, Paula Gaspar, Ángel Muñoz, Miguel Escribano and Elena González is well written and I have very little issue with the paper.

However, one main point I am unable to discern is the difference in gross revenue between indoor and outdoor production. Are the economics of the two systems being compared with an identical market value, or are the differences in value simply differences in market price fluctuation? I do not know much of the Iberian marketing system, but are indoor and outdoor pigs paid the same, or is there a difference with outdoor pigs given a premium because they are outdoor raised?

Dear reviewer, as a clarification to your comment the analysis has been done on real farms and cases. Depending on the type of farm the value of their productions is different. In supplementary material we have included the evolution of the prices of the two systems of exploitation of the Lonja de Extremadura. This shows a higher selling price for pigs raised outdoors, as this characteristic is highly appreciated by the consumer, but also responds to the higher costs of the production model.

An explanatory note has also been included in the text.

It would seem to me that you would want to compare the two systems on an equal market price per kg of pork. If this is the case, then I think the data would suggest an economic advantage to indoor-raised pigs.  If there are premiums to outdoor raised pigs, then this should be added on top of an identical market price. From reading the paper, I can not determine if the systems are being compared on an equal market value basis, or just the arbitrary market prices received at the time pigs happened to be slaughtered.

In response to your question, and as a continuation of the previous one, the data were analyzed according to the real market price. These prices are not arbitrary and respond on the one hand to the extra price granted by the consumer's preferences, but on the other hand what really sets the prices are the production costs of the exploitation systems.

I think the economic aspects of the paper need to be explained and prices equalized before consideration for publication.

Due to the above-mentioned characteristics, it does not make sense to equalize prices for the analysis when we talk about two well differentiated products in the market.

At the same time, the text of the article clarifications has been added in the text.

Reviewer 3 Report

Although this is a study that uses multiple farms, I would consider adding some type of diet composition table so that the readers can determine if the nutrient specs on these farms are similar to others.

While this is a nice study, this is a production style paper and not appropriate for this journal.

Author Response

Response to Reviewer 3 Comments

Although this is a study that uses multiple farms, I would consider adding some type of diet composition table so that the readers can determine if the nutrient specs on these farms are similar to others.

All farms are following the same feeding programme, administering three types of feed (table 2), appropriate to the needs of the pigs.

The feeds are supplied by the same feed company, so that the characteristics in terms of ingredients and nutritional composition are the same throughout the 6 farms.

Round 2

Reviewer 1 Report

The statistics analysis should be revisited. The factor such as the days at farm should be considered including in equation. Moreover the results and discuss post pandemic between two production system is missing. 

CM: Please specify the variable used in model for growth data and economic measure, and have data tested for normality.

Table 2 The decimal place of P value should be adjusted to where number can be read.

CM: The decimal for those traits that found more than 3 place should show as P < 0.001. Probability value can't be zero if that make sense.

Author Response

Response to Reviewer 1 Comments

The statistics analysis should be revisited. The factor such as the days at farm should be considered including in equation. Moreover the results and discuss post pandemic between two production system is missing.

Following the reviewer's comments, the statistical analysis has been revised and the number of days has been added to the equation for the analysis of pig systems. These results are shown in the text both in the methodology part and in the results, Table 4.

CM: Please specify the variable used in model for growth data and economic measure, and have data tested for normality.

Variables used in model have been specified in the text together with information related to normality.  

Table 2 The decimal place of P value should be adjusted to where number can be read.

Corrected.

CM: The decimal for those traits that found more than 3 place should show as P < 0.001. Probability value can't be zero if that make sense.

Corrected.

Reviewer 2 Report

I have no further comments.

Author Response

I have no further comments.

Dear reviewer,

Thank you very much for your contributions.

Best regards.

Reviewer 3 Report

Table 4 is improved.  It now shows a snapshot in time for these two systems.  However, if the purpose of this study is to document the common price differences across the two systems, then an additional table is needed.  In this table the following should be considered: if there are no premiums for raising pigs outdoors compared to indoors, then one should use the same sell price for the pigs.  In addition, one should also use the same purchase price for the pigs.  If the feed ingredients are the same, then feed price per kg should be the same as well.

Author Response

Response to Reviewer 3 Comments

Table 4 is improved.  It now shows a snapshot in time for these two systems.  However, if the purpose of this study is to document the common price differences across the two systems, then an additional table is needed.  In this table the following should be considered: if there are no premiums for raising pigs outdoors compared to indoors, then one should use the same sell price for the pigs.  In addition, one should also use the same purchase price for the pigs.  If the feed ingredients are the same, then feed price per kg should be the same as well.

Dear reviewer, as a clarification to your comment the analysis has been done on real farms and cases. Depending on the type of farm the value of their productions is different. In supplementary material we have included the evolution of the prices of the two systems of exploitation of the Lonja de Extremadura. This shows a higher selling price for pigs raised outdoors, as this characteristic is highly appreciated by the consumer, but also responds to the higher costs of the production model.

In this sense, and as a continuation of the previous one, the data were analyzed according to the real market price. These prices are not arbitrary and respond on the one hand to the extra price granted by the consumer's preferences, but on the other hand what really sets the prices are the production costs of the exploitation systems.

Round 3

Reviewer 3 Report

None